# Micelle Encapsulation of Ferromagnetic Nanoparticles of Iron Carbide@Iron Oxide in Chitosan as Possible Nanomedicine Agent

**Perla Yazmin Sauceda-Oloño [1], Hector Cardenas-Sanchez [2]** ,
**Anya Isabel Argüelles-Pesqueira [2]** , **Cindy Gutierrez-Valenzuela [2],**
**Mario Enrique Alvarez-Ramos [1], Armando Lucero-Acuña [1,2]** and **Paul Zavala-Rivera [1,2,\*]**

[1] Posgrado en Nanotecnología, Departamento de Física, Universidad de Sonora, Hermosillo 83000, Mexico; psaucedao@gmail.com (P.Y.S.-O.); enrique.alvarez@unison.mx (M.E.A.-R.); armando.lucero@unison.mx (A.L.-A.)

[2] Departamento de Ingeniería Química y Metalurgia, Universidad de Sonora, Hermosillo 83000, Mexico; harturo.cardenassanchez@hotmail.com (H.C.-S.); anya.arguelles@unison.mx (A.I.A.-P.); cindy.gutierrez@unison.mx (C.G.-V.);

\* Correspondence: paul.zavala@unison.mx; Tel.: +52-662-182-7787

**Abstract:** In this work, the synthesis and characterization of core/shell nanoparticles of iron carbide@iron oxide ($Fe_3C/\gamma\text{-}Fe_2O_3$) encapsulated into micelles of sodium dodecylsulfate and oleic acid and stabilized with chitosan was developed. The materials were sonosynthesized at low intensities using standard ultrasonic baths with iron pentacarbonyl ($Fe(CO)_5$) and oleic acid as iron source and hydrophobic stabilizer, respectively; obtaining nanoparticles with a hydrodynamic diameter of 19.71 nm and polydispersive index (PDI) of 0.13. The iron carbide@iron oxide nanoparticles (ICIONPs) in oleic acid were used as the organic phase during the self-assemble of nanoemulsion with sodium dodecylsulfate in water to obtain the metastable micelles. The final step involved the stabilization of the micelles using low molecular weight chitosan solution at 2% in acetic acid by ultrasonication bath. The nanosystem showed a hydrodynamic diameter of 185.30 nm, a PDI of 0.15 with a superficial charge $\zeta$ of 36.70 mV. Due to the magnetic, physical and chemical properties previously measured of the ICIONPs, it is believed that this type of nanoparticles can be used as a possible nanomedicine agent.

**Keywords:** nanocapsules; iron carbide; iron oxide; chitosan

## 1. Introduction

Over recent decades, the applications of nanoparticles, especially iron oxide nanoparticles have been widely explored. For example maghemite ($\gamma\text{-}Fe_2O_3$) nanoparticles have shown possible applications in biomedicine due to the biocompatibility tested on in vitro and in vivo studies [1,2]; and also, its magnetic properties that allow their manipulation through external magnetic fields [3]. However, most of its applications are focused in electrophotography, hyperthermia for applications on cancer cell elimination, diagnosis through magnetic resonance, among others [4,5].

The main purpose of the use of magnetic nanoparticles is that because of their size, most of them show superparamagnetic behavior, that makes them useful as contrast agent in MRI diagnosis [5,6]. Besides, the use of ferromagnetic and ferrimagnetic nanoparticles have been exploded for its ability to undergo into hyperthermia induced by high-frequency magnetic fields, causing cellular death when exposed to the thermal effect [7,8]. Coating and encapsulating the nanoparticles helps to obtain intelligent responses, such as: change the capacity of being water or oil soluble;

surface positive or negative charge; and diverse biomedical applications as magnetic resonance [9], hyperthermia therapy [10], drug delivery [11], tissue [12] and genetic engineering [13]. Besides, increase biocompatibility, cell/tissue selectivity, increase the external magnetic fields response and be able to tune physical and chemical properties as desired.

Usually, the preparation of these type of nanocomposites involves the interaction of an organic phase (solvent, stabilizer polymer, organic acid and hydrophobic drug or particle) and an aqueous phase (water, alcohol, surfactants and hydrophilic drug or particle) [14,15]. Once the nanoparticle has been stabilized the different surface modifications done, this could lead to other surface interactions with saccharides, proteins, enzymes or antibodies found in the biological systems. Due to these characteristics, it is needed in a theranostic agent (that works as a hyperthermic therapy and diagnosis agent), a system based in a composite material based on a well-known hyperthermic system ($Fe_3C$) and another one used as contrast agent ($\gamma$-$Fe_2O_3$) were chosen.

In this work, we explore the preparation of nanoencapsulated particles made of iron carbide@iron oxide ferrimagnetic nanoparticles (ICIONPs) synthesized through a novel method based on sonochemistry; the complete synthesis and characterization has been published in previous work [16], in which those particles had demonstrated a ferrimagnetic behavior. These particles were encapsulated using surfactants and chitosan as a stabilizer with the technique of nanoprecipitation, leading to the formation of nanosystems of 6nm within capsules below 300nm; the properties as a ferrimagnetic nanoparticle shows a plausible application as theranostic agent due to its capacity for being used as contrast agent and hyperthermia in presence of high-frequency magnetic fields.

## 2. Materials and Methods

### 2.1. Materials

The following compounds where acquired through Sigma-Aldrich, and used without further purification: iron pentacarbonyl ($Fe(CO)_5$), oleic acid (OA), dioctyl ether, absolute alcohol, toluene, sodium dodecyl sulfate (SDS), acetone, low molecular weight chitosan, glacial acetic acid and ultrapure water. The statistical studies were made using R version 1.70 supplied with R 3.5.2 as R Studio version 1.1.463 for Mac.

### 2.1.1. Iron Carbide@Iron Oxide Nanoparticles (ICIONPs)

For the synthesis of ICIONPs, a previously published methodology without modifications was used [16], in which a dilution of 1 mmol of $Fe(CO)_5$ and 3 mmol of OA in 5 mL of dioctyl ether was sonicated in an ultrasonic bath (Branson 3510) for 30 min. $Fe(CO)_5$ was used as the iron source and the OA as the surface modification in which the dioctyl ether function as a solvent. The solutions were placed on 15 mL tubes with lid which were opened every 10 min to aid oxygen diffusion. During the sonication, a turn in color from yellow to dark brown or black is observed.

The resulting solution was then purified to remove unreacted material and to obtain the most monodisperse family of nanoparticles. During this step, three rounds of centrifugation at $7500\times g$ for 20 min are used. The first step involved the precipitation of the nanoparticles with a ratio that worked with 3 mL of absolute alcohol for every 2 mL of solution. The precipitated nanoparticles were resuspended in 2 mL solution at 12% *v/v* of OA in toluene, after centrifugation the supernatant was collected. In the next step, the supernatant was precipitated again with the same relation of absolute ethanol. Finally, the precipitation was resuspended in 2 mL of the desired organic solvent (usually toluene).

### 2.1.2. Nanoemulsion and Nanoparticles

The methodology was modified from a previous works [17] published by Rosas-Durazo et al. Once the nanoparticles were sonosynthesized, they were dispersed in a 2 mL mixture 1:1 of toluene-OA. ICIONPs were fully disperse in the solution using a ultrasonication bath for 5 min. Then, the mixture

was rotoevaporated at 40 °C under vacuum to remove the corresponded volume of toluene, finally 9.5 mL of acetone and 88 µL of OA are used as previous experiments has shown; this will be known as the organic phase.

For the nanoemulsion formulation, the organic phase was slowly added into a stirred aqueous phase, consisting of different concentrations of SDS in 20 mL of ultrapure water, based on the average value of its critical micelle concentration (CMS). At the moment of the interaction, the formation of a turbid/white solution presenting Tyndall effect should immediately appear.

This solution was rotoevaporated at 40 °C and 80 RPM under vacuum, for about 30 min or until a volume of approximately 10 mL was left. This solution was saved for further characterization and nanoencapsulation.

### 2.1.3. Nanocapsules Fabrication

With the metastable emulsion, the nanocapsules were produced by the precipitation with the addition of 5 mL of acetic acid solution at 2% and 2.5 mg of low molecular weight chitosan. After 30 min of continuing stirring, a 100 µL volume of the past solution was diluted in 9.9 mL of ultrapure water which was left in ultrasonication bath for 5 min. The final solutions were kept stored at 4 °C until further use.

### *2.2. Characterization*

### 2.2.1. Scanning and Transmission Electron Microscopy

A TEM model JEOL JSM-7800F and a STEM model JEOL JEM-2010F were used in the morphological characterization of the nanoparticles and nanocapsules. The TEM was used at 200 keV and STEM using 30 keV of acceleration. The samples were prepared on carbon on 400 mesh copper grids (Ted Pella Inc.), stained with phosphotugsting acid and left to dry at vacuum overnight.

### 2.2.2. Dynamic Light Scattering and Laser Doppler Electrophoresis

The hydrodynamic diameter and Z potential ($\zeta$) of the ICIONPs, the emulsion droplets and the chitosan coated micelles were measured using a Zen5600 nanoseries from Malvern Instruments with a light source of He-Ne of 633 nm at a temperature of 37 °C. In the case of the ICIONPs, toluene was used as solvent while ultrapure water was used for the micelles and chitosan nanocapsules. Refractive index of oleic acid was used for all the measurements, which were made by triplicate. The pH of the chitosan coating nanoparticles were neutralized with HCl and NaOH both at 0.01 M.

### 2.2.3. FTIR Spectroscopy

Fourier Transform Infrared spectra was made on a Nicolet iS50 from Thermo Scientific using ATR mode with diamond window. The parameters used were: resolution of 2 $cm^{-1}$, 64 scans with autogain from 4000 to 500 $cm^{-1}$; the samples with organic solvent were dry at vacuum and the aqueous based were lyophilized, both left overnight. All the samples were prepared and measure by triplicate.

## 3. Results and Discussions

### *3.1. Scanning and Transmission Electron Microscopy*

The TEM micrographs (Figure 1a) showed the expected core@shell structure, in which, as previous work has shown, the core is made of iron carbide ($Fe_3C$) and the shell from iron oxide($\gamma$-$Fe_2O_3$) [16]; with an average size of 13.60 ± 3.63 nm (n = 501).

The difference between nanocapsules with and without nanoparticles can be observed in the micrographs of Figure 1b,c from scanning electron microscopy, showing size from 213.80 ± 12.65 nm (n = 35) to particles diameters of 291.30 ± 10.97 nm (n = 27). The micrographs show ideal spherical shape with a difference within a bigger diameter of around 100 nm from nanocapsules without ICIONPs;

this difference could be explained by the steric effect of the nanoparticles within the oleic nucleus of the micelles causing an alteration on the final size.

Finally, using TEM images, the presence of the magnetic nanoparticles within the nanocapsules was characterized as shown in Figure 1d. In the picture, the presence of cumulus of nanoparticles inside the nanocapsules can be easily observed, specially how it starts to precipitate inside of the oleic nucleus.

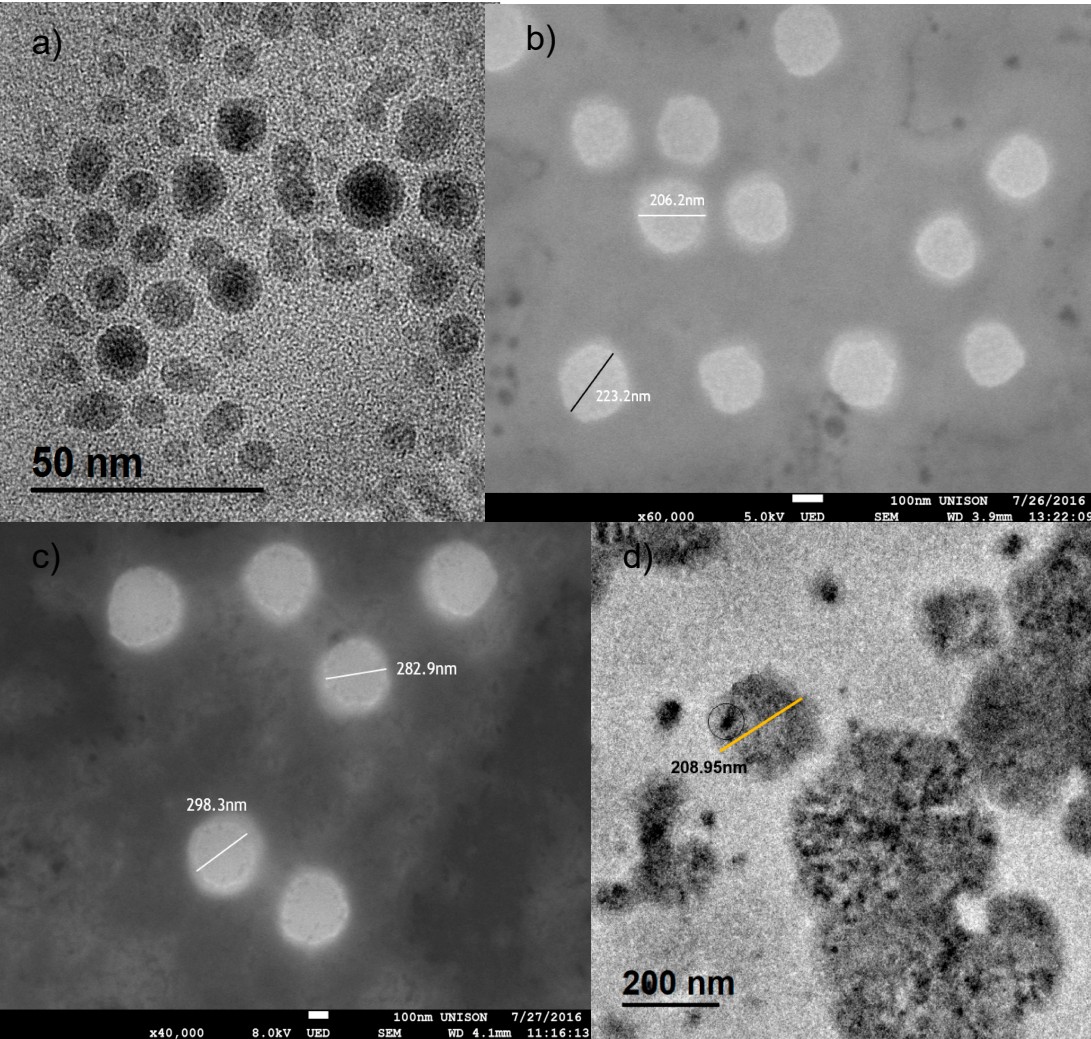

**Figure 1.** TEM and SEM micrographs of the iron carbide@iron oxide nanoparticles and the iron carbide@iron oxide chitosan coated nanocapsules. (**a**) TEM micrograph showing the average size of 13nm with a nucleus of iron carbide and a shell of iron oxide. (**b**) SEM micrograph of chitosan coated nanocapsules without nanoparticles. (**c**) SEM micrograph chitosan coated nanocapsules with nanoparticles within. (**d**) TEM micrograph of chitosan coated nanocapsules with nanoparticles within.

## 3.2. Dynamic Light Scattering

Figure 2 shows the different measurement of the hydrodynamic diameter and the PDI obtained by the algorithm within the Malvern software. Figure 2a, shows an average size of 19.71±1.36 nm and a PDI of 0.13 for the ICIONPs used in further experiments; this size difference against the TEM results are easily explained trough the hydrodynamic effect of the oleic acid coating around the solid nanoparticle.

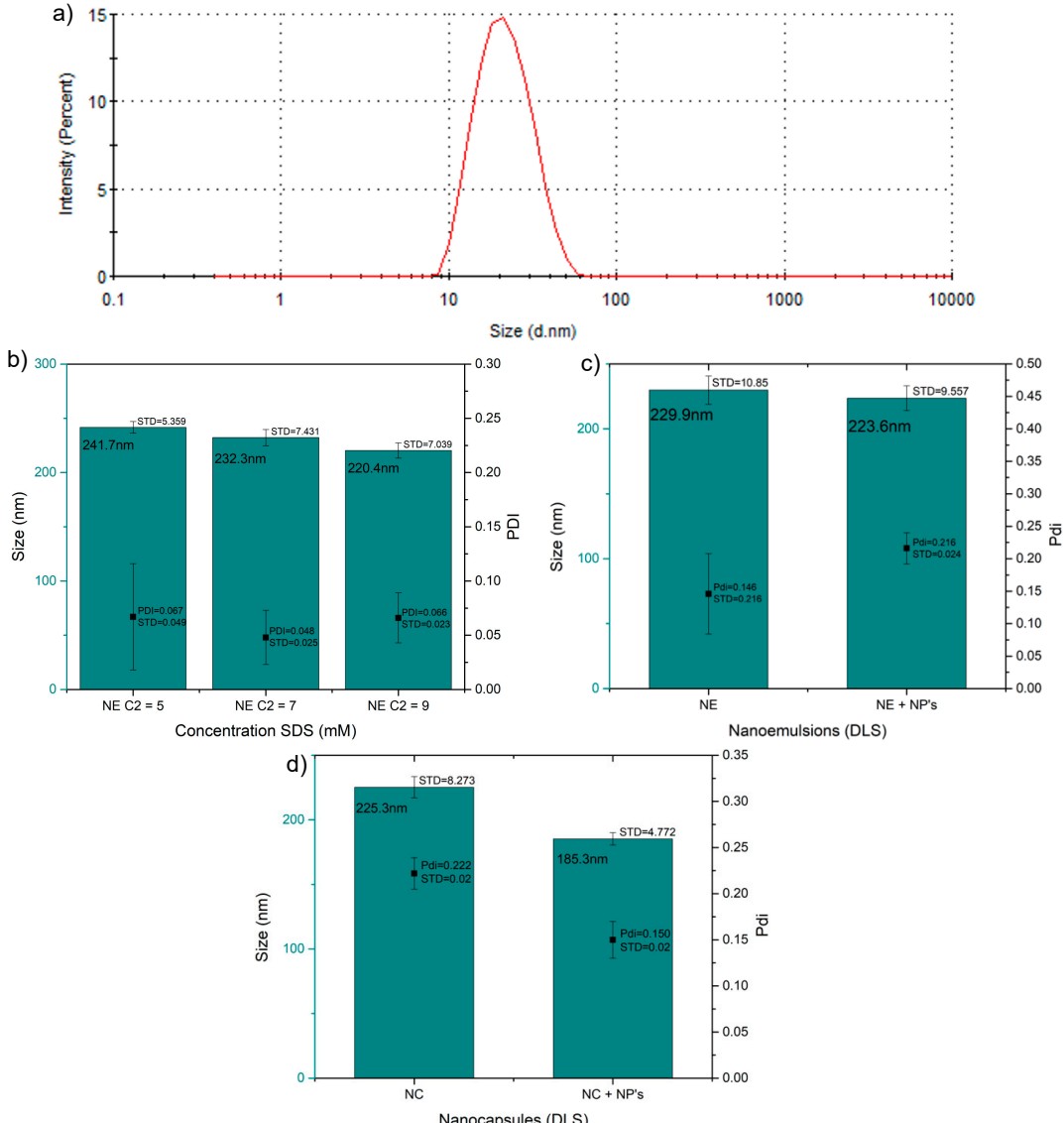

**Figure 2.** Hydrodynamic diameter values of the nanoparticles, emulsions and capsules. (**a**) Hydrodynamic diameter distribution of the iron carbide@iron oxide nanoparticles (NP´s) in toluene. (**b**) Hydrodynamic diameter and PDI of the uncoated nanoemulsion (NE) without nanoparticles at different concentrations. (**c**) Hydrodynamic diameter and PDI of the uncoated nanoemulsions with and without ICIONPs within the micelles. (**d**) Hydrodynamic diameter and PDI of chitosan coated nanocapsule (NC) with and without ICIONPs within.

Using the critical micelle concentration (CMC) of SDS as standard, different concentration of SDS (5, 7 and 9 mM) were made to observed the effect of SDS concentration on the size and PDI of the emulsions, the concentration with the minimal size and metastability was chosen (Figure 2b). It was found that statistically, the minimum significant size and polydispersity was obtained when using a SDS concentration of 7 mM, giving and average hydrodynamic diameter of 232.30 ± 7.43 nm and PDI of 0.05 ± 0.03. This configuration was used in further experiments.

In Figure 2c it is shown that using this concentration (7mM SDS), the effect in the size and PDI of the micelles were studied and compared with the nanoparticles within them; with the purpose of observing the effect of them in the final size; a statistical difference was not found, and both systems showed similar hydrodynamic diameter around 225.30 and a PDI of around 0.18 ($p > 0.1$).

The effect of the iron carbide@iron oxide nanoparticles in the hydrodynamic diameter size and PDI of the chitosan coated nanocapsules was studied (Figure 2d). Overall, these results provide some

evidence of the emulsion and coating achievements during the different steps of the encapsulation; adding the final coating did not change significantly the final size of the hydrodynamic diameter, keeping an average size of 225.30 ± 8.27nm, the same as the nanoemulsion shown in Figure 2c. The unexpected behavior was found after the encapsulation of chitosan of the nanoemulsion loaded with the nanoparticles, in which was observed a decrease of the overall nanocapsule diameter to 185.30 ± 4.77 nm ($p < 0.001$). This change could be explained by the effect of precipitation of the nanoparticles inside the micelle oleic nucleus.

### 3.3. Laser Doppler Electrophoresis

The $\zeta$ of the samples was measured by triplicate and at temperatures of 37 °C and pH equal to 7 ± 1.3. Figure 3a shows the $\zeta$ of the nanoemulsion produced by the different concentration of SDS with expected negative values; concentration of 7 and 9 mM did not show a statistical significant difference, while the concentration of 5 mM it shows an expected reduction in its surface charge from around −42.40 ± 1.12 mV to −40.30 ± 0.32 mV due to the concentration of SDS on its surface ($p < 0.001$). Furthermore, the addition of the iron carbide@iron oxide nanoparticles did not show any significant effect in the $\zeta$ (Figure 3b); keeping both emulsion with a surface charge around −43.50 mV, the statistical analysis hasn´t shown a statistical difference with a $p = 0.017$, although the excess of oleic acid could minimally affect the zeta potential of micelles with the nanoparticles inside. After coating the emulsions with chitosan, the $\zeta$ rise to the positive side of the charge as showed in Figure 3c without a noticeable change when in presence of the ICIONPs; keeping a surface charge of +36.30 mV, showing not a statistical significance expected for the type and concentration of chitosan coating ($p > 0.1$).

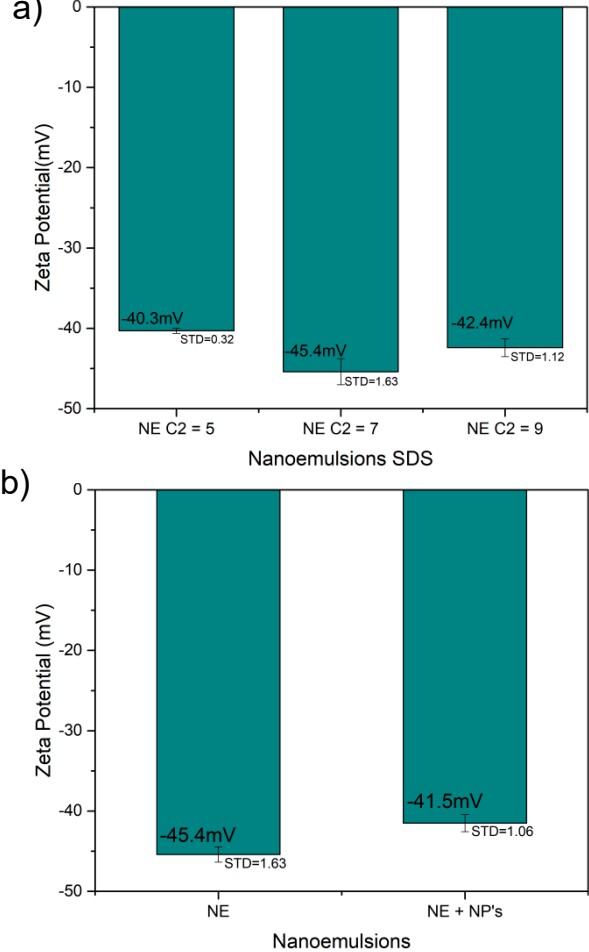

**Figure 3.** *Cont.*

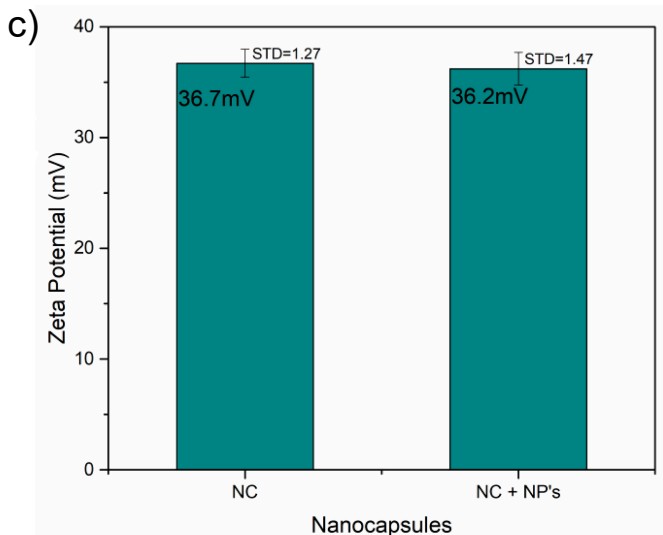

**Figure 3.** Zeta potential plots of the (**a**) emulsions, (**b**) the nanoparticle effect on the emulsions and (**c**) the nanocapsules with and without the nanoparticles.

### 3.4. FTIR Spectra

During the characterization, changes in the functional groups were searched through the comparison of the different precursor before and after the emulsion (Figure 4a), the nanoparticle addition (Figure 4b) and the chitosan coating during nanoencapsulation (Figure 4c).

For the emulsion, the interaction between SDS, OA and ICIONPs was searched. SDS showed bands around 3465 cm$^{-1}$ corresponding to O-H stretching, while at the nanoemulsion it is found around the band of 3298 cm$^{-1}$. At 2956 cm$^{-1}$ the asymmetric vibration of methyl in the SDS can be localized, as well the symmetric and asymmetric stretching of methylene in the alkane chain −CH2− at 2916 and 2849 cm$^{-1}$, respectively. The bands of the symmetric and asymmetric stretching of the S=O group were found at 1215 and 1079 cm$^{-1}$, respectively, as well as in the nanoemulsion where they were found with a displacement in the bands of 1235 and 1067 cm$^{-1}$.

Oleic acid showed its characteristic bands of asymmetric and symmetric stretching from is methylene groups at 2923 and 2852 cm$^{-1}$, proper from is aliphatic chain. The intense band at 1706 cm$^{-1}$ correspond to the stretching of the C=O from the carboxylic group. Signals found in the pure substance as well as in the nanoemulsion.

The comparison between the nanoemulsion with and without ICIONPs showed the presence of the band corresponding to both oleic acid and SDS. Besides, a new band at 1006 cm$^{-1}$ which correspond to the stretching of the group C=O found in the nanoparticles due to the interaction between the oleic acid and the iron oxide [16].

Finally, the chitosan coating was observed looking through the characteristic bands like the O-H stretching at 3315 cm$^{-1}$, the C−O stretching of the amide I at 1652 cm$^{-1}$ and the glycosidic link C−O−C at 1026 cm$^{-1}$; these bands were also found in the nanocapsules spectra.

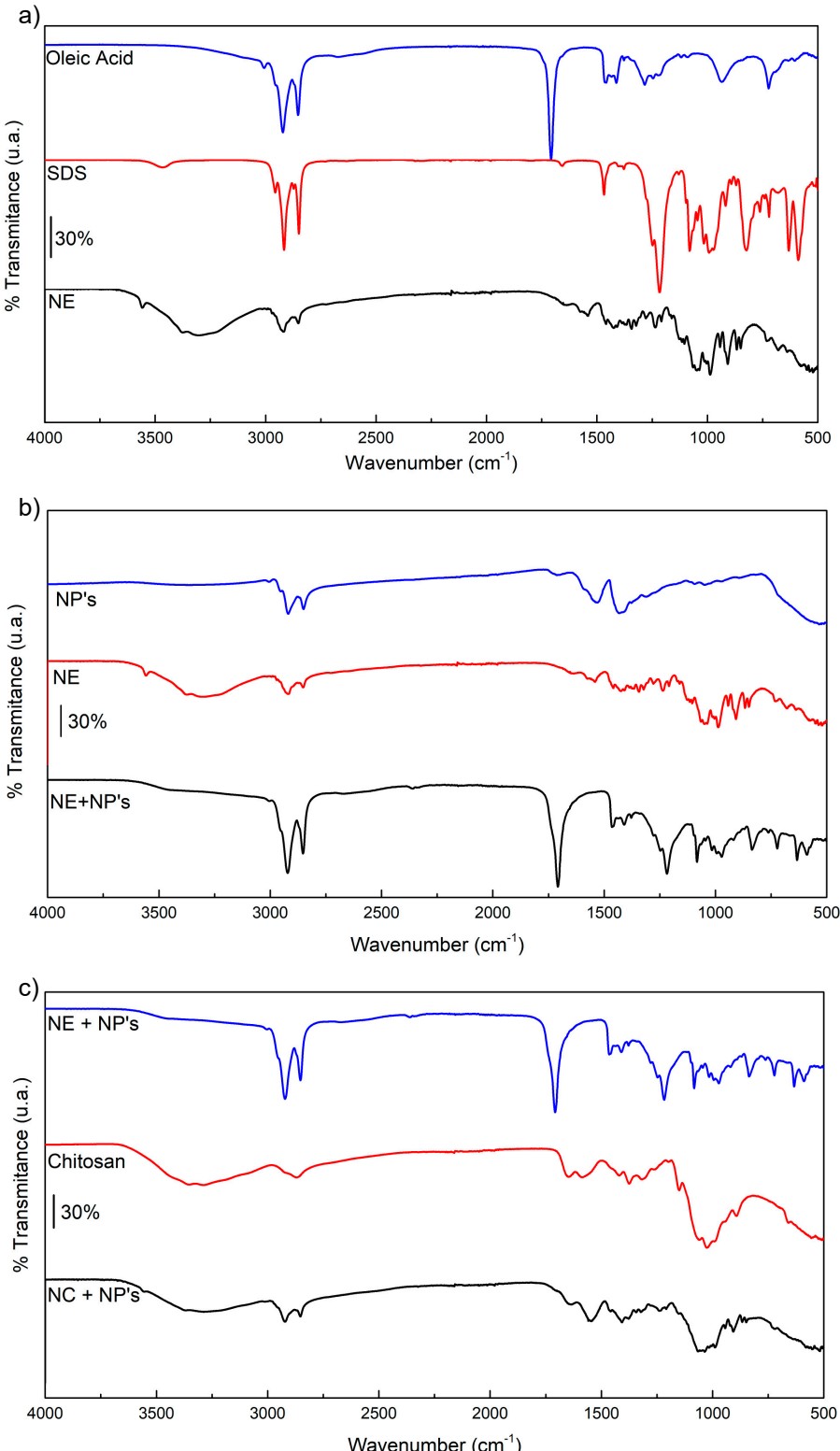

**Figure 4.** FTIR spectra of the (**a**) nanoemulsion (NE) and its precursors: oleic acid and sodium dodecyl sulfate (SDS). (**b**) Nanoemulsion with and without iron carbide@iron oxide nanoparticles (NP′s). (**c**) Nanocapsules (NC) and their precursors: chitosan and nanoparticles. The bars indicate 30% intensity of transmittance.

## 4. Conclusions

The results presented in this study clearly demonstrate the encapsulation of iron carbide@iron oxide nanoparticles through a two-step process: (i) nanoparticles were sonosynthesized using low intensity ultrasonication baths with an average dry diameter found through TEM of 13.60 nm and hydrodynamic around 19.70 nm; (ii) the nanoparticles were suspended in oleic acid and sodium dodecyl sulfate at a concentration of 7 mM to create a nanoemulsion with hydrodynamic diameter of 235.30 nm and a zeta potential of −41.50 mV; (iii) nanoemulsion was coated with low molecular weight through a nanoprecipitation process, demonstrated by its change in the zeta potential from a negative charge to a positive one of +36.30 mV and hydrodynamic radius of 185.50 nm. This diminution in the size could be explained due to the instability of the ICIONPs in the oleic nucleus.

Accordingly to the FTIR spectrometry, chemical reactions through the different steps where not found beyond the expected one between $Fe(CO)_5$ and oleic acid at the moment of synthesize the iron oxide magnetic nanoparticles.

Finally, the nanoencapsulation of iron oxide magnetic nanoparticles opens a possibility of using this system as theranostic agent. Similarly to nanosystems which find the diseases [18] and by external or magnetically effects aid a therapy [19]. Furthermore, the chitosan coating opens the possibility of its use because of its biocompatibility properties mentioned herein. Furthermore its zeta potential is adequate to a good stability if its compare with similar systems [20,21].

**Author Contributions:** Conceptualization, P.Z.-R. and A.L.-A.; methodology, P.Z.-R., A.I.A.-P., H.C.-S. and P.Y.S.-O.; formal analysis, P.Y.S.-O. and M.E.A.-R.; investigation, P.Y.S.-O., H.A.C.-S and A.I.A.-P.; resources, P.Z.-R., A.L.-A. and M.E.A.-R.; data curation, P.Z.-R., H.C.-S. and C.G.-V.; writing—original draft preparation, P.Y.S.-O. and P.Z.-R.; writing—review and editing, P.Z.-R. and C.G.-V.; visualization, A.I.A.-P.; supervision, P.Z.-R.; project administration, P.Z.-R.; funding acquisition, P.Z.-R., M.E.A.-R. and A.L.-A. All authors have read and agreed to the published version of the manuscript.

**Funding:** This research was funded by "Secretaria de Educación Pública" SEP, México, grant number UNISON-PTC-250 and the "Consejo Nacional de Ciencia y Tecnología" CONACYT, México through the "Becas nacionales" program.

**Acknowledgments:** P.Z.-R. sincerely appreciate the significant contributions made by Eduardo Larios-Rodríguez in the TEM in Tranmission Electron Microcopy Laboratory from the Universidad de Sonora and Roberto Carillo-Torres in the STEM characterization of the samples.

**Conflicts of Interest:** The authors declare no conflict of interest.

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
