# Peer review of "Micelle Encapsulation of Ferromagnetic Nanoparticles of Iron Carbide@Iron Oxide in Chitosan as Possible Nanomedicine Agent"

_colloids, doi:10.3390/colloids4020022_

Round 1

Reviewer 1 Report

The manuscript needs thorough editing at many points, the structure needs changes, e.g. the experimental methods are partially presented among the results, but some technical detail is still missing. The figures and figure captions also need editing to help the readers in better understanding. All the chapters should be improved, there are grammatical and technical errors too. The conclusions are not supported by the results presented in the text, since there were no experiments (e.g. magnetic measurements, MRI or magnetic hypertehrmia tests, biocompatibility studies, etc.) carried out to to test/prove the newly snythesized NPs' theranostic potential.

Comment and questions:

Title: „of iron carbide@iron oxide ferromagnetic....”

Line 12-13: same correction as in the title

Line 16-17: „These nanoparticles were used as the organic phase…” – how can be a NP the organic phase?

Line 17-18: what happenned in the final step? something is missing in the sentence

Line 20-21 (last sentence): the synthesis of NPs is only the first step - to apply a NP for theranostics, the biocompatibility and other properties should be checked

Line 31-32: the main purpose is that they show superparamegnetism?

Line 36-37: „is, .... to obtain, ... to change...., to give...., to provide the appropriate feature for...”

Line 38-39: „the different phases” it is not necessary

Line 40-41: in some cases the surface modification increases the biocompatibility too, they are not separate processes

Line 44-45: why iron carbide? why core-shell NPs? why sonochemistry? no any words were about these before

Chapter 2: experimental techniques used for characterization are missing

miliQ water: Milli-Q® is a trademark, it is better to use the term ultrapure water instead

Line 60-61: what does AO stands for?

Line 68-69: why those volumes (not integer, but fractional values) were used? are they based on earlier experience or calculated values?

Line 75: mixture instead of solution

Line 72, 82, 93: these sentences belong rather to the results

Line 85-86: last sentence; should be mentioned as the first sentence of the paragraph

Line 89: too much nano and this is not necessary; emulsion and precipitation is enough without nano prefix

Figure 1 caption: Line 117-118: only the name of the samples should be mentioned here; C and D can be merged, since both are mentioning the same kind of the sample only the imaging technique is different („….TEM (C) and SEM (D)”)

Line 123: what does the term „nanoemulsion particles” mean? particles is (nano)emulsion or emulsion droplets should be used instead

Line 161: hydrodynamic diameter instead of DLS characterization

Figure 2: PDI instead of Pdi; the same number of digits should be used for STD as it was used for the data (e.g. 253.2 ± 0.2 nm)

Line 168: the name of the method is laser Doppler electrophoresis....

Figure 3: it is better to show the data in the first two graphs in the opposite direction, because they are negative; it is enough to give the STD with one decimal point, because the ZP is also given in that format

Line 183: ZP or zeta potential

Fig.4: please align the graphs and use the same size for each one; what does the 30 % stand for on the graph? a, b and c is missing from the graphs

Line 217: FTIR spectra instead of FTIR spectroscopy

Line 223: sonosynthesized

Line 224: accessible - why it is so important?; by instead of through

Line 225: one decimal is enough

Author Response

First of all, I would like to thank you for the opportunity and time invested in reviewing our work; all you comments have been invaluable in making this work a lot better.

Starting with your fist recommendation, all the figures were revised, checked and fixed. About the comment on the conclusion, we modified the title and conclusion to a broad spectrum, because the iron carbide@iron oxide nanoparticles synthesis and characterization has been published previously, showing a ferrimagnetic behavior and chemical characteristics.

The use of the "@" in the title and else, is because most papers explaining core-shell colloids and nanoparticles used them. e.g.

Synthesis and Self-Assembly of Au@SiO2 Core−Shell Colloids. Nano Letters 2002, 2, 7, 785-788

Finally, all your grammar and editing has been take into account, as well as it has proof read by someone with advanced english level.

And again, thanks again for you comments!

Reviewer 2 Report

The paper presents a simple way to make these nanoparticles, followed by a traditional (but comprehensive) characterization effort. I would suggest the following modifications:

  • The paper talks about theranostics, but there is no discussion of how these particles would fit into that landscape. Why do authors think that these NP could make a difference? Any chance to test them over any cell line?
  • Do they know the magnetic susceptibility? 
  • Figures are poorly formatted in the sense that the colors/fonts sizes selected will make them hard to read once published. There is no point in writing the values inside the bars (Fig 3)
  • I would recommend reformatting Fig 3A to match the others 
  • I would recommend that they include the error bars in the discussion and potentially perform some statistical analysis to justify that particles are (or not) similar
  • This reviewer is not sure about the meaning of "pH close to 7". This should be verified. 

Author Response

Dear Reviewer,

First of all, I would like to thank you for your time. Everyone of your comments has been take into account to make this work better.

As you recommended all the figures has been modified and checked, as well transformed in a more easy to use digital format.

The text has been checked, statistical analysis has been employed, and error bars has been used in the text. The value of the pH has been verify as you pointed out.

Finally, you comments on magnetic characterization has been done in the previous publication showing a ferrimagnetic behavior. This has been clarify in the text and an explanation on why this nanoparticle is different has been explained. Also, testing the nanoparticles encapsulated them in chitosan are going to be tested in a near future, hoping the pandemia crisis left us come back to the laboratory.

And again, Thanks a lot for your time.

Round 2

Reviewer 1 Report

The manuscript improved after the authors made some changes, but there are still some questions and comments which were not considered during the revision process:

  1. Description of an experimental technique belongs rather to the Methods section, please move them into the corresponding part (2.2.) of the manuscript.
  2. Line 49-50: still no any word in the Introduction why did the authors choose the iron@iron carbide system. What are the advantages or drawbacks of this material? What are the challenges of their biomedical use? Please add some information about it to establish the state-of-art better.
  3. The use of the numbers for presenting any value (e.g. size) should be uniform why are the STDs given with three decimals? If the ZP is given with an accuracy of one decimal digit, it is enough for the STD too. Please use the same number of decimals throughout the manuscript (for all the other data as well).
    E.g. one digit is enough, 42.4 ± 1.1 mV instead of 42.4 ± 1.12 mV.

Line 46-47: in some cases, the surface modification and the stabilization takes place together, the stabilization is a consequence of the changes made on the NP’s surface.

Line 76-77: why these volumes? Are the calculated values (or empirical data)? If yes, how did you determined?

Fig. 2: if the size date were obtained by Zen5600 device, the raw data can be exported from the software (if the right PSD tab is active, using the Edit/Copy Size Values command will copy the data into the clipboard and from there they can be inserted (and edited) into Excell or Origin etc.

Line 161: “characterization” is not necessary, maybe “values” can be used instead

Line 169-170: only electrophoretic mobilities can be measured, the ZP can be calculated from these data using e.g. the Smoluchowski equation. The “pH is around 7” or “pH is equal to 7.0 ± 1.3”, please decide which form to use.

Fig. 3: it would be nice to use the same column width in each graph.... (part b)

Fig. 4: this is a spectra of something, not spectroscopy; this latter is a name of a method. Please use e.g. “Infrared spectra” instead.

Author Response

Again, Thanks for your time took to review our work. The different points will be address herein:

1. Description of an experimental technique belongs rather to the Methods section, please move them into the corresponding part (2.2.) of the manuscript.

A.- Sorry, I could find the section that you mentioned, any experimental technique described outside the methodology were sample preparation for its use in the different equipments.

2. Line 49-50: still no any word in the Introduction why did the authors choose the iron@iron carbide system. What are the advantages or drawbacks of this material? What are the challenges of their biomedical use? Please add some information about it to establish the state-of-art better.

A.- Changes were made in Line 50 - 54, but most of the drawback and challenges will be address in future works.

3. The use of the numbers for presenting any value (e.g. size) should be uniform why are the STDs given with three decimals? If the ZP is given with an accuracy of one decimal digit, it is enough for the STD too. Please use the same number of decimals throughout the manuscript (for all the other data as well). 
E.g. one digit is enough, 42.4 ± 1.1 mV instead of 42.4 ± 1.12 mV.

A.- Change it

4. Line 46-47: in some cases, the surface modification and the stabilization takes place together, the stabilization is a consequence of the changes made on the NP’s surface.

A.- Change it

5. Line 76-77: why these volumes? Are the calculated values (or empirical data)? If yes, how did you determined?

A.- Most of these data were obtained by empirical experimentation, by adding or using volumes below certain value until the most optimal was found. Beside, other volumes or concentration were use without changes from previous publications.

6. Fig. 2: if the size date were obtained by Zen5600 device, the raw data can be exported from the software (if the right PSD tab is active, using the Edit/Copy Size Values command will copy the data into the clipboard and from there they can be inserted (and edited) into Excell or Origin etc.

A.- Sadly, The data was obtained from an equipment in other university, in which their user send the data like a Print screen from the software. Also, the pandemic time has keep close the access to the equipments and the users.

7. Line 161: “characterization” is not necessary, maybe “values” can be used instead

A.- Change it

8. Line 169-170: only electrophoretic mobilities can be measured, the ZP can be calculated from these data using e.g. the Smoluchowski equation. The “pH is around 7” or “pH is equal to 7.0 ± 1.3”, please decide which form to use.

A.- Change it, but it was left like that by recommendation fro the other reviewer.

9. Fig. 3: it would be nice to use the same column width in each graph.... (part b)

A.- Change it

10. Fig. 4: this is a spectra of something, not spectroscopy; this latter is a name of a method. Please use e.g. “Infrared spectra” instead.

A.- Change it

Finally, I really appreciate your comments and make our work a better piece.

Regards

Reviewer 2 Report

Authors have addressed my initial questions. 

Author Response

Thanks, I really appreciate the time invest on our work to make it better.

Kind Regards!

Round 3

Reviewer 1 Report

Thank for the authors for considering my comments and replying to my questions. I accept all the answers, but there is still one task left as I asked before:

According to the journals recommendations regarding the "Manuscript Preparation", the descriptions of the experimental details (materials' synthesis, applied techniques) should be collected and described in the "Materials and methods' section, not in the "Results and discusssion".

The current 2.2.1.-2.2.3. sections belong rather to the „Materials”, and should be renumbered to 2.1.1.-2.1.3.. Beside this, the 2.2. Section should be dedicated to the experimental descriptions (e.g. 2.2.1. Scanning and transmission electron microscopy) like it is given in lines 108-108; 134-138; 173-174 and 195-197 according to the journals recommendation in the "Instructions for Authors/Manuscript preparation" tab.

Author Response

Thanks again,

All the recommendation were taking into account in the last modified version.

Kind Regards!